# The Influence of the Toxic Dinoflagellate *Alexandrium minutum*, Grown under Different N:P Ratios, on the Marine Copepod *Acartia tonsa*

**DOI:** 10.3390/toxins15040287

**Published:** 2023-04-16

**Authors:** Epaminondas D. Christou, Ioanna Varkitzi, Isabel Maneiro, Soultana Zervoudaki, Kalliopi Pagou

**Affiliations:** 1Hellenic Centre for Marine Research (HCMR), Institute of Oceanography, P.O. Box 712, 19013 Athens, Greece; ioanna@hcmr.gr (I.V.); tanya@hcmr.gr (S.Z.); popi@hcmr.gr (K.P.); 2Istituti di Ricerca per l’Ambiente Marino, Consiglio Nazionale delle Ricerche (IAS CNR), SS di Capo Granitola, Via del Mare 3, Torretta Granitola, 91021 Campobello di Mazara, Italy; isabelmaneiro@gmail.com

**Keywords:** *Alexandrium minutum*, harmful algal blooms, HABs, *Acartia tonsa*, PSP toxin, toxic algae, PST, toxic microalgae

## Abstract

HABs pose a threat to coastal ecosystems, the economic sector and human health, and are expanding globally. However, their influence on copepods, a major connector between primary producers and upper trophic levels, remains essentially unknown. Microalgal toxins can eventually control copepod survival and reproduction by deterring grazing and hence reducing food availability. We present several 24-h experiments in which the globally distributed marine copepod, *Acartia tonsa*, was exposed to different concentrations of the toxic dinoflagellate, *Alexandrium minutum*, grown under three N:P ratios (4:1, 16:1 and 80:1), with the simultaneous presence of non-toxic food (the dinoflagellate *Prorocentrum micans*). The different N:P ratios did not affect the toxicity of *A. minutum*, probably due to the low toxicity of the tested strain. Production of eggs and pellets as well as ingested carbon appeared to be affected by food toxicity. Toxicity levels in *A. minutum* also had an effect on hatching success and on the toxin excreted in pellets. Overall, *A. minutum* toxicity affected the reproduction, toxin excretion and, to an extent, the feeding behavior of *A. tonsa.* This work indicates that even short-term exposure to toxic *A. minutum* can impact the vital functions of *A. tonsa* and might ultimately pose serious threats to copepod recruitment and survival. Still, further investigation is required for identifying and understanding, in particular, the long-term effects of harmful microalgae on marine copepods.

## 1. Introduction

In recent years, harmful algal blooms (HABs), due to their increased frequency and duration, have turned into a threat to marine ecosystems, causing serious ecological and economic issues [1,2,3]. Zooplankton is a major component of the marine pelagic food web, both by directing primary production to higher trophic levels and by regulating phytoplankton populations (top-down control). More specifically, zooplankton grazing plays a key role in controlling phytoplankton blooms, as well as those of harmful species [4]. Algal blooms occur when phytoplankton growth is higher than zooplankton grazing. Hence, HABs are believed to be assisted by reduced grazing due to the algae generating toxic matters [5]. However, it has also been suggested that grazing pressure is not a major mechanism favoring toxic over non-toxic strains before bloom initiation [6]. Xu et al. [7] proposed that the various *Alexandrium* strain-aversion responses of copepods may have very different impacts on the toxic species and their capacity to form blooms.

The impacts of toxic phytoplankton species on zooplankton grazers, ranging from feeding suppression and physiological depletion to decreases in egg production and hatching success, have been well documented [8]. In particular, the adverse effects of toxic blooms on copepod growth and reproduction have been reported in several field and laboratory investigations, e.g., [9,10,11]. However, it has not been clear if the reduced consumption of toxic cells is the result of discriminatory feeding or possibly a physiological reaction to consumed toxins. In general, investigated grazer responses to toxic phytoplankton species are diverse. Concerning copepods, the dominant zooplankton group, their responses to harmful algae may differ within and among species of both the algae and grazers, and responses may vary from uninhibited to important [12]. Recently, Trapp et al. [13] showed that the history of copepod abundance correlates with increased toxin levels in farmed mussels, whereas selective-feeding- and grazer-instigated toxin production may explain this pattern in some toxic dinoflagellates. They also suggested that measuring the increase in copepodamides, a group of copepod-produced compounds triggering toxin formation, could be an innovative and low-cost approach for including previous grazing intensity in new techniques about understanding and predicting HABs. However, at present, it is actually impossible to predict, even roughly, how copepods will respond to an upcoming HAB.

Among toxic algae, the dinoflagellate genus *Alexandrium*, having caused significant destruction to aquaculture, is among the most investigated toxic dinoflagellates [3]. The genus *Alexandrium* is a well-known Paralytic Shellfish Toxin (PST) producer. PST production and other physiological responses of several *Alexandrium* species have been shown to vary with the source, disposal and amount of dissolved inorganic nitrogen and phosphorous [14]. Mostly, PST concentration is directly related with nitrogen concentration [15,16]. The available source of nitrogen is a regulatory factor for PST levels since ammonium has been found to induce significantly higher toxins than when cultured under nitrates or urea as the nitrogen source for some *Alexandrium* spp. (*A. catenella*, *A. excavatum*, and *A. tamarense*) [17,18]. Furthermore, the lack of phosphorous has been found to enhance toxin levels of *Alexandrium* spp. (*A. minutum* and *A. tamarense*) when compared to N-replete or N-depleted cells [14,19]. In the case of *A. minutum* in particular, N:P ratios and the different available nitrogen compounds have been shown to affect the growth of *A. minutum* as well as the cellular toxin percentage and the toxin production rate [16]. Actually, the cellular toxin quota increased under phosphorus limitation (with high N:P ratios) in both nitrate and ammonium treatments [16]. In particular, for the toxin production rate, this was enhanced under phosphorus limitation in ammonium-supplied cultures, whereas there was no difference detected in nitrate or ammonium-supplied cultures when phosphorus was replete [16].

The effects on copepods exposed to *Alexandrium* spp., nevertheless, may vary from no impact to negative impacts on the ingestion rate, egg production, egg hatching and offspring development duration [20,21,22,23,24]. Roncalli et al. [23], investigating the impact of the toxic dinoflagellate *Alexandrium fundyense* on the copepod *Calanus finmarchicus*, reported that blooming of *A. fundyense* in the Gulf of Maine had an adverse effect on copepod recruitment, whereas the consumption of *A. fundyense* did not affect its survival [24]. It was finally demonstrated in what way deep-sequencing technology can interpret numerous mechanisms of toxin resistance at the same time, enlightening the linkages between molecular/cellular adaptations and *C. finmarchicus* ecology [24].

Concerning *A. tonsa*, it has been indicated that adult females can obtain enough energy from the toxic *Alexandrium catenella* to survive, but undergo reproductive deficiency when consuming this prey alone [25]. Breier and Buskey [26], studying the impact of the red tide dinoflagellate *Karenia brevis* on the grazing and fecundity of *A. tonsa*, suggested that *K. brevis* is not harmful to *A. tonsa*. However, it lacks some chemical constituent responsible for inducing a feeding response in *A. tonsa* along with the nutritional requirements for normal reproduction. Lasley-Rasher et al. [27] highlighted the necessity to identify how grazing on HAB species modifies consumer behavior, as this may have major effects on the fate of toxins in marine ecosystems.

In the present study, we tested the effects of the toxic dinoflagellate *Alexandrium minutum* on the commonly co-occuring copepod *Acartia tonsa*. We performed several 24-h experiments in which the globally distributed marine copepod, *A. tonsa*, was exposed to different concentrations of the toxic dinoflagellate *A. minutum*, grown under three diverse N:P ratios, with the simultaneous presence of non-toxic food (the dinoflagellate *Prorocentrum micans*). This work supplements the existing information on the toxin fate through copepods and also indicates the influence of toxins on vital copepod functions.

## 2. Results

### 2.1. Toxin Impact on Ingestion of A. tonsa

The cultures of *A. minutum* were grown under three different N:P treatments: nitrogen-limited (4:1), phosphorus-limited (80:1) and N:P-replete (16:1) conditions. Overall, the cell toxin contents ranged from 1.74 to 6.07 fmol cell^−1^ (as saxitoxin equivalents); and after a comparison among the three different N:P treatments, the toxin levels per cell were rather similar (on average, 3.29 fmol cell^−1^ ± 1.69 SD). On these grounds, three pools of results were created, stemming from three food toxicity levels—L = low, M = medium and H = high—in relation to *A. minutum* cell concentrations.

At the different *A. minutum* concentrations (total of the experiments), ingestion increased with an increase in concentration (one-way ANOVA, F = 24.40, *p* < 0.0001); and was higher at 2000 and 3000 than at 500 and 1000 cell ind^−1^ d^−1^ (LCD test, *p* < 0.05; Figure 1a). Concerning *P. micans* (always at 100 cell ind^−1^ d^−1^), ingestion also increased with an increase in *A. minutum* concentration (one-way ANOVA, F = 14.10, *p* < 0.0001); and was higher at 2000 and 3000 than at 0, 500 and 1000 cell ind^−1^ d^−1^
*A. minutum* (LCD test, *p* < 0.05).

Figure 1b shows the ingestion of *A. minutum* for the three toxicity levels (L = low, M = medium, and H = high) in relation to *A. minutum* cell concentrations. A multiple regression model (stepwise variable selection) on *A. minutum* ingestion rates as a dependent variable (cells ind^−1^ d^−1^, *Ale_I*) and various independent variables {*A. minutum* cell concentrations (*Ale_C*), *P. micans* cell concentrations (*Pro_C*), cell toxicity, food carbon concentration} revealed that *A. minutum* and *P. micans* concentrations are the predictors of *A. minutum* ingestion rates according to the equation:log *Ale_I* = 1.037 − 0.662*log *Pro_C* + 0.797*log *Ale_C* (r^2^ = 0.55, F = 24.08, *p* < 0.01)

An ANCOVA, taking *A. minutum* ingestion as the dependent variable, food toxicity level as a factor and *A. minutum* and *P. micans* concentrations as covariables, showed that *A. minutum* cell concentration was significantly correlated with the ingestion rate (F = 18.05, *p* < 0.001), whereas toxicity level had no effect, confirming the multiple regression model.

A simple regression between ingested carbon (*IC*) and cell toxicity (*CT*) revealed that carbon ingestion decreased with increasing cell toxicity, according to the equation: log *IC* = 4.073 − 0.767*log *CT*, (r^2^ = 0.40, F = 20.82, *p* < 0.001).

Finally, the toxicity level did not reveal any significant effect on the ingestion rate. However, the increasing *A. minutum* food concentrations increased ingestion of both *A. minutum* and *P. micans*, indicating an ability to actively select non-toxic cells when in parallel ingesting toxic cells, hence reflecting a minor impact on the feeding behavior of *A. tonsa*. In addition, carbon ingestion decreased with increasing cell toxicity.

### 2.2. Toxin Accumulation in Tissues of A. tonsa

At the different *A. minutum* concentrations (total of the experiments), toxin accumulation in tissues, ranging between 183 and 588 fmol ind^−1^, an increase in the concentration (one-way ANOVA, F = 17.27, *p* < 0.0001); and was higher at 3000 than at 500, 1000 and 2000 cell ind^−1^ d^−1^ (LCD test, *p* < 0.05; Figure 2a).

A simple regression between accumulated toxin in body (*AT*) and toxin ingestion (*TI*) revealed that toxin accumulation was positively dependent on toxin ingestion, according to the equation: log *AT* = 1.062 + 0.793*log *TI*, (r^2^ = 0.32, F = 24.38, *p* < 0.05). In Figure 2b, accumulated toxin by copepods seems to be lower at the medium toxicity levels. However, an ANCOVA, taking toxin ingestion as a covariable and food toxicity level as a factor, indicated that there were no significant differences among toxicity levels (F = 13.07, *p* < 0.1). Hence, in contrast to our expectations, no effect of the food toxicity level on the accumulation of toxins by copepods was confirmed. Perhaps our data were quantitatively inadequate to support such a significant relationship. However, a significant increase in toxin accumulation in tissues was confirmed at 3000 cell ind^−1^ d^−1^ (Figure 2a).

### 2.3. Egg Production of A. tonsa

At the different *A. minutum* concentrations (total of the experiments), egg production increased at the beginning and then decreased with the increase in concentration (one-way ANOVA, F = 9.99, *p* < 0.0001); and was higher at 500 and 1000 than at 0, 2000 and 3000 cell ind^−1^ d^−1^ (LCD test, *p* < 0.05; Figure 3a).

Figure 3b shows egg production of *A. tonsa* in relation to carbon ingestion at the three toxicity levels (L, M, and H). An ANCOVA, taking ingested carbon as a covariable and toxicity level as a factor, revealed that there are significant differences between the toxicity levels (F = 24.09, *p* < 0.005).

Regressions between egg production (*EP*) and ingested carbon (*IC*) for each toxicity level showed that egg production increased with carbon ingestion rates at low toxicity levels according to the equation:log *EP* = 1.243 + 0.441*log *IC*, (r^2^ = 0.71, F = 17.44, *p* < 0.005)

However, egg production decreased as carbon ingestion increased at high toxicity levels according to the equation: log *EP* = 4.391 − 0.834*log *IC*, (r^2^ = 0.43, F = 12.08, *p* < 0.005).

No effect was detected at the medium toxicity level. Therefore, the high *A. minutum* concentrations and the high toxicity levels significantly decreased egg production.

### 2.4. Egg Hatching of A. tonsa

At the different *A. minutum* concentrations (total of the experiments), egg hatching decreased with the increase in concentration (one-way ANOVA, F = 10.24, *p* < 0.0001). This appears to be a gradual decrease; however, it was higher at 0 than at 1000 and 2000 cell ind^−1^ d^−1^, at 500 than 2000 cell ind^−1^ d^−1^ and at 1000 than 2000 cell ind^−1^ d^−1^ (LCD test, *p* < 0.05; Figure 4a).

Hatching success appeared to be lower at high toxicity levels (Figure 4b). An ANCOVA, taking toxin ingestion as a covariable and food toxicity level as a factor, revealed significant differences among toxicity levels (F = 6.13, *p* < 0.01), with hatching reduced at high toxicity levels. However, no effect of toxin ingestion on hatching success was confirmed.

A simple regression between hatching success (*H*) and accumulated toxin by copepods (*AT*) confirmed the reduced hatching success with an increase in accumulated toxin according to the equation:*H* = 182.695 − 26.101*log *AT*, (r^2^ = 0.15, F = 4.84, *p* < 0.05).

Therefore, hatching success significantly decreases when, either the *A. minutum* concentrations increases, or the toxin accumulated in copepod tissues increases.

### 2.5. Fecal Pellet Production of A. tonsa

At the different *A. minutum* concentrations (total of the experiments), fecal pellet production did not show any significant change with the increase in concentration (one-way ANOVA, F = 0.71, *p* < 0.59, Figure 5a).

According to Figure 5b, at high toxicity and high ingestion, pellet production decreased significantly, whereas there is a positive relationship between pellet production and ingestion rates at low and medium toxicity.

An ANCOVA, taking ingested carbon as a covariable, food toxicity level as a factor and pellet production as a dependent variable, confirmed that there are significant differences between toxicity levels (F = 9.51, *p* < 0.001), and that pellet production was affected by ingested carbon (F = 13.49, *p* < 0.001).

Regressions between pellet production (*PP*) and ingested carbon (*IC*) for each toxicity level showed that pellet production increased with carbon ingestion at the low toxicity level {log *PP* = 1.536 + 0.617*log *IC*, (r^2^ = 0.89, F = 56.04, *p* < 0.001)} and at the medium toxicity level {log *PP* = −1.012 + 1.077*log *IC*, (r^2^ = 0.52, F = 7.65, *p* < 0.05)}. At the high toxicity level, no more pellets are produced when carbon ingestion increases.

The different *A. minutum* concentrations did not significantly affect fecal pellet production. However, high toxicity levels had a significant strong negative impact on pellet production.

### 2.6. Excreted Toxins in Pellets of A. tonsa

At the different *A. minutum* concentrations (total of the experiments), toxin accumulation in pellets increased with an increase in the concentration (one-way ANOVA, F = 8.06, *p* < 0.0003) and was higher at 1000 than at 500 cell ind^−1^ d^−1^ (LCD test, *p* < 0.05; Figure 6a).

Figure 6 shows the toxin excreted in pellets for the three toxicity levels, in relation to toxin ingestion. An ANCOVA, taking toxin ingestion as a covariable and food toxicity level as a factor, indicated differences among toxicity levels (F = 14.39, *p* < 0.001). Hence, toxin excretion in pellets was significantly different at the three toxicity levels (H > L > M), according to our expectations.

## 3. Discussion

PST concentrations are positively correlated with nitrogen concentration [15,16], as the nitrogen source can be a regulatory factor for PST levels [17,18]. In addition, phosphorous limitation has been found to enhance toxin levels of *Alexandrium* spp. when compared to N-replete or N-depleted cells [14,19]. Concerning *A. minutum*, N:P ratios and the available nitrogen compounds have been reported to control its growth as well as the cellular toxin percentage and the toxin production rate [16]. Different levels of nitrogen have been applied in previous works, using high concentrations of nitrogen (100–500 µM) in experimental schemes [28,29], while other authors used lower nitrogen concentrations [30,31]. 

In the present study, *A. minutum* cultures were grown in three different N:P treatments: nitrogen-limited (4:1), phosphorus-limited (80:1) and N:P-replete (16:1) conditions. However, the cell toxin contents were found to be rather similar among the different N:P treatments we tested (on average, 3.29 fmol cell^−1^ ± 1.69 SD) and this can be attributed to the rather low cell toxin content of the specific *A. minutum* strain used. For example, a number of diverse toxin profiles described for *A. tamarense* have been suggested to be related with the phylogenetic variations among different strains of *A. tamarense*, although more detailed studies are yet to be conducted in order to thoroughly confirm the relationship between phylogeny and toxin profiles [32,33,34].

Copepods are reported to be adversely affected when feeding on toxic dinoflagellates, with symptoms dependable on toxicity [20,35,36,37] and decreased fitness [38,39]. However, toxic dinoflagellates were ingested by copepods without apparent negative impacts, in other cases [40,41]. It has been found that when the copepod *A. clausi* was exposed to increasing concentrations of *A. minutum*, in a mixed diet of toxic and non-toxic cells, copepods preyed actively on the toxic *A. minutum* with no signs of decreased ingestion or fullness by the toxic cells [21,22]. Guisande et al. [21] also stated that the grazing capacity of the copepod *A. clausi* feeding on toxic *A. minutum* was not affected by the ingestion of toxins.

In the present study, it was shown that *A. tonsa* fed actively on *A. minutum* and that the ingestion rate increased at higher food concentrations. At the same time, the feeding demand on the non-toxic dinoflagellate *P. micans* also increased, although its concentration did not change. Within this context, we also found that carbon ingestion decreased with increasing cell toxicity. A study, carried out with *A. minutum*, *P. micans* and the copepod *A. clausi*, reported that when the toxin content was low, copepods fed mostly on *A. minutum*; but as it became higher, feeding pressure was greater on the non-toxic *P. micans* and lower on *A. minutum* [21]. Assuming that *A. tonsa* is more resistant to the *A. minutum* toxin than *A. clausi*, this feeding strategy appears to be comparable with our study, where the feeding pressure was increased on *P. micans* but did not decrease on *A. minutum*.

Abdulhussain et al. [41] confirmed that *A. tonsa* can concurrently consume both toxic and non-toxic cells, but enhances clearance rates to non-toxic food as the relative concentration of the toxic *Alexandrium catenella* increases. Its capacity to energetically select non-toxic cells at the same time as ingesting toxic cells suggests that the intake of the toxic food does not cause physiological incapability and therefore it has no impact on *A. tonsa* ingestion. However, Colin and Dam [42], using a mixed diet, demonstrated that a highly toxic strain of *Alexandrium* reduced the total ingestion rate of *A. tonsa*, as the proportion of *Alexandrium* increased in the diet. In the above studies, the cell toxin content obviously played a key role.

In our study, toxin accumulation was positively correlated with toxin ingestion, as well as, a significant increase in toxin accumulation in tissues was confirmed at the highest *A. minutum* concentration. Hamasaki et al. [43] reported that during a bloom of *Alexandrium tamarense* in Hiroshima Bay, the amount of toxins preserved by copepods was a function of *A. tamarense* cell density and cellular toxicity. They also found that the copepod *Acartia omorii* stored a much higher amount of PSP toxins in the lab than in the field. Teegarden et al. [44], attempting to form the toxin budgets for three common copepods feeding on the toxic *Alexandrium fundyense*, showed that the occurrence of alternate food did not significantly modify the efficacy of toxin retention and that total toxin amount retained and retention efficacy were different among the three copepods. Concerning toxin accumulation by *A. clausi* fed on *A. minutum*, it has been reported that there were no differences in toxins accumulated in tissues between the single and the mixed diet, indicating that the cell number of the ingested toxic strain was similar for both diets [6].

We found that at the highest *A. minutum* concentrations and at the high toxicity level, the egg production rate decreased significantly. In general, the negative effects of toxic blooms on copepod reproduction have been reported in several field and laboratory studies, e.g., [9,10,11]. Concerning toxic dinoflagellates, several studies suggest that copepod fecundity is adversely influenced by ingestion [20,42,45], whereas other studies report no impact of toxic dinoflagellates on egg production [21]. We propose that the documented variability of cell toxin content among different *A. minutum* strains and different experimental manipulations concerning tested toxicities could be the key aspect in the above discrepancies.

On the other hand, *A. tonsa* can select among different *Alexandrium* species with different toxicities, preferring the least toxic strains [35,41]. It has also been suggested that copepods consume toxic cells at lower rates than non-toxic cells of a similar size [35]. It is therefore thinkable that, in the present study, the decreased egg production at the high toxicity level might be triggered by starvation [25]. Another explanation could be that toxins reduced food absorption efficiency and carbon assimilation; and, consequently, decreased the amount of energy available for egg production [25].

The negative effects of toxic blooms on copepod egg viability have been reported in several studies, e.g., [10,11,39]. We have found decreased hatching success with an increase in *A. minutum* concentration, indicating diet inadequacy or/and a possible reproductive toxicity (i.e., a harmful impact on reproduction functions). As we also found a decreased hatching rate with increasing toxin accumulated in copepod tissues, reproductive toxicity is suggested as the main cause for the decreasing hatching rate. In fact, Guisande et al. [22], studying the fate of PSP toxin ingested by the copepod *A. clausi*, stated that a small percentage (0.98%) of the assimilated toxins was forwarded to the eggs, and this amount increased with an increase in ingestion. They also proposed that, although this small amount has no impact on toxin fate in copepods, this has a major impact on copepod life cycle, as a low egg hatching rate followed the elevated toxin accumulation in copepod tissues. Hence, the explanation that the harmful influence of toxins deposited in eggs may account for the reduced hatching success, proposed in other similar studies [21,22], appears to be well fitted in the present study too.

In our study, at the high toxicity level, when carbon ingestion increased, a significant strong negative impact on fecal pellet production of *A. tonsa* was recorded, indicating that, after a threshold, as the toxicity level increases, pellet production slows down. The only explanation relies on the possible differences in assimilation efficiencies, due to the presence of toxins, finally reflecting overall reduced food assimilation. Except for food concentration, food quality has also been reported to be included among the major factors controlling fecal pellet production in copepods [46,47]. Frangoulis et al. [47], studying the bloom impact of the toxic dinoflagellate *Dinophysis acuminata* on the copepod *A. clausi*, stated that the *D. acuminata* bloom had no impact on the fecal pellet production of the dominant copepod *A. clausi*.

Sedimentation of copepod pellets consists of a major contribution of matter to the benthic fauna and could be an alternate process of toxin transport to the inhabitants of the seabed. Coprophagous species in the pelagic food web can also consume pellets, thus making toxins available to other organisms, which do not directly consume harmful algae. In this study, we found that toxin excreted in pellets increases as the toxicity level increases. Nevertheless, the toxin content in the fecal pellets represent only a low fraction of the toxins ingested by the copepods, or when compared with those accumulated in the copepod tissues [22]. Concerning the present study, the significance of this path, as a link transferring toxins to the upper trophic levels, consequently seems to be rather limited. In a coastal ecosystem, indeed a fraction of the carbon sequestered by the HABs is directly consumed by zooplankton and is also transferred directly to the seabed. However, our results indicated that HABs do not have a major impact on the carbon cycle or on zooplankton populations.

## 4. Conclusions

The different N:P ratios did not affect the toxicity of the dinoflagellate *Alexandrium minutum*, probably due to the low toxicity of the tested strain. However, the availability and toxicity of *A. minutum* as a food source for *Acartia tonsa* appeared to influence the vital functions of the copepod and, to a lesser degree, its feeding behavior. In this context, copepod ingestion rates did not differ significantly; however, carbon ingestion decreased with increasing cell toxicity. Egg production and hatching decreased at high toxicity levels, as well as, hatching success was reduced when toxin accumulation increased in copepods. As the toxicity level increased, pellet production slowed down and more toxin was excreted in pellets at higher cell toxicity levels (H > L > M).

Overall, *A. minutum* toxicity affected the carbon ingestion, egg production, hatching success, pellet production, toxin excretion and, to a lower extent, the feeding behavior of *A. tonsa*. It appears that, even short-term exposure to toxic *A. minutum* can affect the vital functions of *A. tonsa*, which might pose serious threats to copepod recruitment and survival in the long run. Still, further investigation is required for identifying and understanding the long-term effects of toxic microalgae on marine copepods. It is also evident that additional research is necessary for understanding how the energetic and nutritional needs of *A. tonsa* are influenced by ingesting toxic species.

## 5. Materials and Methods

### 5.1. Experimental Design

*Alexandrium minutum* cultures (strain CCMP 113) were grown in f/2 medium under 3 different N:P ratios (4:1, 16:1 and 80:1). *Prorocentrum micans* cultures (strain CCAP 1136/15) were cultivated in f/2 medium under balanced N:P conditions 16:1. Females of *Acartia tonsa* were grown in laboratory cultures with a supply of *Isochrysis galbana* and *Rhododmonas salina*. All cultures were kept at 20 °C, salinity 30 and under a 16:8 h light–dark cycle. In all treatments we attempted to maintain the concentration of *P. micans* at approximately 100 cells ml^–1^, whereas the concentrations of *A. minutum* were 0, 500, 1000, 2000 and 3000 cells ml^–1^. The carbon concentration in all treatments including *A. minutum* was sufficient to certify that food selection was not affected by potential food restriction, which might force copepods to ingest less palatable cells [48]. All copepods were starved for 48 h before the experiment. Two individuals of *A. tonsa* were transferred to each vial (25 vials: 20 replicates and 5 controls for each treatment) containing 53 mL of the different food concentrations. Each incubation lasted for 24 h in all experiments. The cell concentration of *A. minutum* and *P. micans*, ingestion rates, egg production, fecal pellet production and egg hatching rates, as well as the toxin amount per cell, toxin amount per copepod and toxin amount per fecal pellet, were estimated. A replicate was excluded if any of the copepods was either dead or moribund.

### 5.2. Vital Rate Estimations

In total, 15 to 20 replicates (2 copepods per 53 mL vial) were used for assessing the ingestion rates of *A. tonsa* on *A. minutum* and *P. micans* for each treatment. Ten replicate control vials and five initial vials without animals were set up concurrently. Samples from the initial vials were kept directly at the beginning of the experiment. After 24 h incubation, copepod mortality was checked and the samples were fixed for phytoplankton cell estimation under an inverted microscope. Cell concentrations were determined by counting in 1 ml in a Sedgewick–Rafter chamber. Ingestion rate calculation was based on the equations produced by Frost [49].

For each treatment, eggs and fecal pellets produced by 35 to 40 individuals were collected. The eggs and pellets produced by 8 to 10 copepods were combined, so that 4 replicates corresponded to each treatment. From each of these replicates, 150 to 200 eggs were incubated for an additional 48 h period and the total nauplii with the remaining eggs were subsequently counted after fixation with formalin. Finally, after checking copepod mortality, the number of fecal pellets and the number of females were also counted.

### 5.3. Toxin Analysis

To estimate the paralytic shellfish poisoning (PSP) cell toxin of *A. minutum*, cells were collected on pre-combusted 13 mm GF/F Whatman filters, kept at −30 °C in ultracentrifuge plastic tubes, and lyophilized. After the addition of 0.05 M acetic acid (500 μL), the sample was homogenized and then frozen. Finally, the material was placed at 4000 rpm for 10 m (twice), after which 200 μL of the supernatant was collected and kept at −30 °C.

For the analysis of the toxin accumulated in copepod tissues, 15 individuals were transferred from each treatment to filtered seawater and, to assure that the total of toxin content was expelled, animals were moved to 40 μL distilled water after 2.5 h and directly collected. The fecal pellets were moved from each treatment to distilled water, and were directly collected in 500 μL distilled water. The pellets from each treatment were joined into one sample, hence the final pellet number varied from 410 to 1610. Samples of copepods and pellets were kept at –30 °C and then the same procedure, as described above, was followed.

PSP toxins (in *A minutum* cells, copepod tissues and pellets) were examined using high-performance liquid chromatography with fluorescence detection (HPLC-FD) following the method of Oshima et al. [50], with modifications by Franco and Fernández [51]. Chromatographic profiles were shaped using quadruplicate injections of 35 μL extracts, diluted with 0.05 M acetic acid. Toxin standards (decarbamoyl saxitoxin), purchased from the National Research Council of Canada (Halifax), were used for the HPLC-FD analysis.

The toxicity of *A. minutum*, in saxitoxin equivalents (STXeq), was estimated from the HPLC chromatograms. For final toxicity determination, toxin concentration values were multiplied by a toxin-specific conversion factor. The specific toxicity conversion factors of the toxins were implemented by Oshima [52], based upon empirical mouse bioassay data determined using purified standards.

### 5.4. Statistical Analysis

For statistical analysis, StatGraphics [for the analysis of variance (ANOVA) and multiple and simple regressions, applied on log-transformed data] and SPSS [for the analysis of covariance (ANCOVA)] were used. Multiple comparisons among the different concentrations of *A. minutum* (five treatments; 0, 500, 1000, 2000 and 3000 cell ml^−1^ in the total of the experiments) concerning all investigated parameters were performed using one-way ANOVA. When significance was reached by the ANOVA results, Fisher’s least significance difference (LSD) test was used to examine whether the means were significantly different among the treatments. Results were accepted as significant at the *p* = 0.05 level.

An ANOVA was performed using the observed cell toxicity as the dependent variable and two factors—the first one with each kind of experimental bottle (initial, control or replicate with copepod) and the second one with the date and nutrient condition. Results indicated that there are differences among date and nutrient condition but not among bottles (F = 4.75, *p* < 0.001). This means that, in each experiment, we could use the same toxicity for initial, control and copepod bottles, which is crucial to minimize errors in statistics. Hence, in each experiment, the same toxicity (mean) was used for each type of experimental bottle. In order to study the possible effects of total food toxicity on copepod feeding and reproductive behavior, the data regarding initial food toxicity were grouped into three toxicity levels: low (L, <2000 fmol ml^−1^), medium (M, 2000–4000 fmol ml^−1^) and high (H, >4000 fmol ml^−1^).

The approach followed in the analysis of each examined parameter (toxin impact on ingestion, toxin accumulation in tissues, egg production, hatching success, fecal pellet production, and excreted toxins in pellets) was based on the consumer response (a) according to the different food concentrations and (b) at the three toxicity levels.

## Figures and Tables

**Figure 1 toxins-15-00287-f001:**
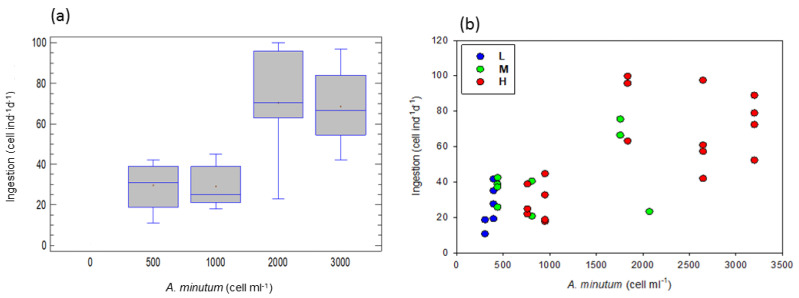
Ingestion rates of *A. minutum*: (**a**) at the different food concentrations, total of the experiments (significantly different; 2000, 3000 > 500, 1000 cell ind^−1^ d^−1^), (**b**) as a function of *A. minutum* concentration. L, M and H are the low, medium and high toxicity levels.

**Figure 2 toxins-15-00287-f002:**
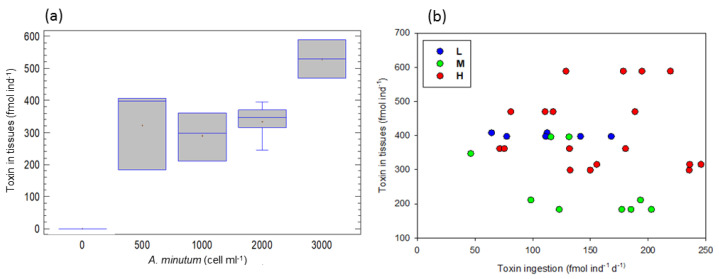
Toxin accumulated in the copepod tissues (**a**) at the different food concentrations, total of the experiments (significantly different; 3000 > 500, 1000, 2000 cell ind^−1^ d^−1^), (**b**) as a function of toxin ingestion rates. L, M and H are the low, medium and high toxicity levels.

**Figure 3 toxins-15-00287-f003:**
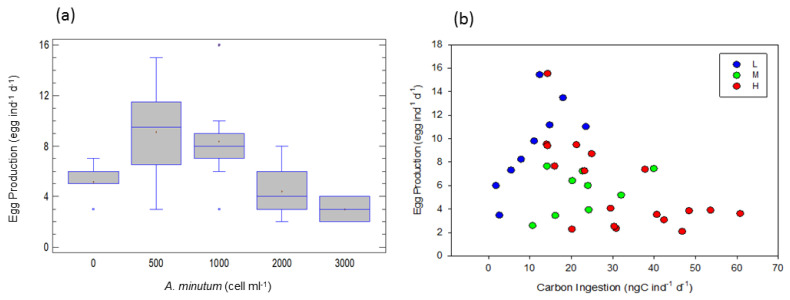
Egg production of *A. tonsa* (**a**) at the different food concentrations, total of the experiments (significantly different; 2000, 3000 < 500, 1000 cell ind^−1^ d^−1^), (**b**) as a function of carbon ingestion rates. L, M and H are the low, medium and high toxicity levels.

**Figure 4 toxins-15-00287-f004:**
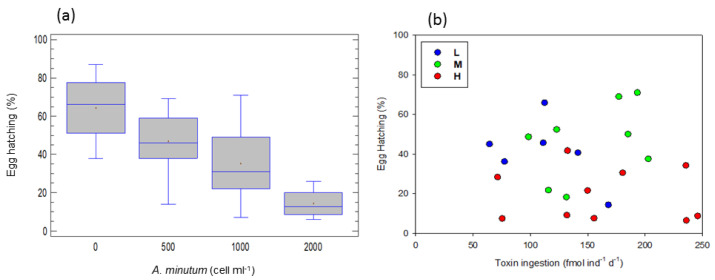
Egg hatching success (%) (**a**) at the different food concentrations, total of the experiments (significantly different; 2000 < 1000, 500, 0 cell ind^−1^ d^−1^, 1000 < 0 cell ind^−1^ d^−1^). The data for 3000 cell ind^−1^ d^−1^ are missing; no egg was hatched at 3000 cell ind^−1^ d^−1^ (**b**) as a function of toxin ingestion rates. L, M and H are the low, medium and high toxicity levels.

**Figure 5 toxins-15-00287-f005:**
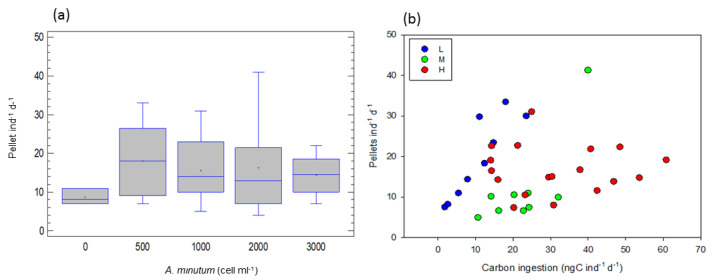
Fecal pellet production of *A. tonsa* (**a**) at the different food concentrations, total of the experiments (lack of any significant difference) (**b**) as a function of carbon ingestion rates. L, M and H are the low, medium and high toxicity levels.

**Figure 6 toxins-15-00287-f006:**
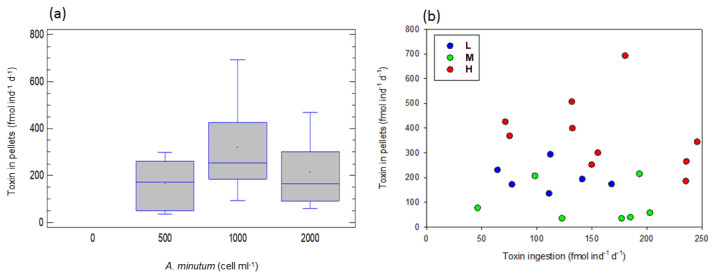
Toxin excreted in the copepod fecal pellets (**a**) at the different food concentrations, total of the experiments (significantly different; 1000 > 500 cell ind^−1^ d^−1^). The data for 3000 cell ind^−1^ d^−1^ are missing; some samples were treated improperly and the number of the remaining ones could not support a reliable measurement. (**b**) As a function of toxin ingestion rates. L, M and H are the low, medium and high toxicity levels.

## Data Availability

Deidentified data are available on request from the corresponding author.

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
