# Peer review of "The Influence of the Toxic Dinoflagellate Alexandrium minutum, Grown under Different N:P Ratios, on the Marine Copepod Acartia tonsa"

_toxins, 2023, doi:10.3390/toxins15040287_

Round 1
Reviewer 1 Report
The manuscript “The Influence of the Toxic Dinoflagellate Alexandrium minutum, grown under different N/P ratios, on the Marine Copepod Acartia tonsa” follows that microalgal toxins can eventually affect copepod survival and reproduction by deterring grazing and hence reducing food availabilty. Although the topic of interactions between toxin-producing microalgae and zooplanktonic organisms that consume them is not very new, it is interesting to treat it from the relationship of nutrients. However in my opinion the most important “bump”, is the issue of “why nutrients do not affect”, seems very vague. This part should be strengthened in the discussion of the manuscript and it is not included in the abstract of the paper either. In general, I do find this manuscript being overall very well written; the statistic is used correctly.The authors might consider a final proofing paying special attention that some numerical value should be shown in the abstract and some references update them (1994, 1996)
Author Response
The manuscript “The Influence of the Toxic Dinoflagellate Alexandrium minutum, grown under different N/P ratios, on the Marine Copepod Acartia tonsa” follows that microalgal toxins can eventually affect copepod survival and reproduction by deterring grazing and hence reducing food availabilty. Although the topic of interactions between toxin-producing microalgae and zooplanktonic organisms that consume them is not very new, it is interesting to treat it from the relationship of nutrients. However in my opinion the most important “bump”, is the issue of “why nutrients do not affect”, seems very vague. This part should be strengthened in the discussion of the manuscript and it is not included in the abstract of the paper either.
There is a comment in the discussion about that (“this can be attributed to the rather low cell toxin content of the specific A. minutum strain used”) with some related literature information. Now this is also included in both the abstract and the conclusions.
In general, I do find this manuscript being overall very well written; the statistic is used correctly.The authors might consider a final proofing paying special attention that some numerical value should be shown in the abstract and some references update them (1994, 1996).
Some numerical values are now included in the abstract. Concerning references, they are already updated, but some older ones are necessary too.

Reviewer 2 Report
Dear Editors and authors
Happy day
Concerning the paper: The Influence of the Toxic Dinoflagellate Alexandriu m minutum, grown under different N/P ratios, on the Marine Copepod Acartia tonsa.
The paper can be significantly improved.
My comment include the following points.
1- In analysis the toxicity of something against a living organism, it is recommended to use the Probit analysis to determine the toxicity levels (e.g., LC50. LC10 etc). You have the data and you can calculate a clear point indicate the toxicity. Yes, you do it indirectly and show decrease due to the toxicity for different parameters. Kindly be more specific. By chance, if you could not find a suitable software, you can calculate it manually using the Probit paper. So, I suggest to keep the result you represent and use the Propit analysis in addition.
2- You need to adjust many of the scientific names (italic and the other standard criteria).
3- I appreciate highly your honest in representing the data but in a sentence like " (in two cases the 3000 cells ml–1 were excluded)" kindly explain why?
4- It is nice to represent the graphs, but why not to add a table to summarize your finding. You can do that against the different treatment and even you can put the (*) on the significant concentration as you find in the statistical analysis part.
5- I have downloaded other papers published in toxin to be sure from this point. Why you did not use subtitles in the material and methods part. Kindly improve.
6- You have an excellent introduction which give a good image but you need to follow each idea or information with at least one reference.
7- L, M and H are the low, medium and high toxicity levels collectively give a hint but did not represent a specific point where one could say at the point x there is a risk. Kindly use the propit analysis.
With my pleasure

Author Response
Concerning the paper: The Influence of the Toxic Dinoflagellate Alexandriu m minutum, grown under different N/P ratios, on the Marine Copepod Acartia tonsa.
The paper can be significantly improved.
My comment include the following points.
1- In analysis the toxicity of something against a living organism, it is recommended to use the Probit analysis to determine the toxicity levels (e.g., LC50. LC10 etc). You have the data and you can calculate a clear point indicate the toxicity. Yes, you do it indirectly and show decrease due to the toxicity for different parameters. Kindly be more specific. By chance, if you could not find a suitable software, you can calculate it manually using the Probit paper. So, I suggest to keep the result you represent and use the Propit analysis in addition.
Many thanks for this suggestion. I have carefully checked the relevant literature and I found that probit analysis has not been used in this kind of experimental works. In general I know that probit analysis is mostly used in standard toxicity tests checking mortality level in organisms exposed to a toxic substance. I cannot think an application way in the experiments of the present study.
2- You need to adjust many of the scientific names (italic and the other standard criteria).
They were adjusted.
3- I appreciate highly your honest in representing the data but in a sentence like " (in two cases the 3000 cells ml–1 were excluded)" kindly explain why?
This sentence was removed, and explanations were included in the captions of the corresponding graphs
4- It is nice to represent the graphs, but why not to add a table to summarize your finding. You can do that against the different treatment and even you can put the (*) on the significant concentration as you find in the statistical analysis part.
Thanks for this idea. The lack of uniformity in the approach (anova, simple regressions, multiple regressions, ancova) makes the summarizing on a table difficult and complex, as well as a kind of repetition (some results are already included in the graph captions).
5- I have downloaded other papers published in toxin to be sure from this point. Why you did not use subtitles in the material and methods part. Kindly improve.
Subtitles were used.
6- You have an excellent introduction which give a good image but you need to follow each idea or information with at least one reference.
It was adjusted.
7- L, M and H are the low, medium and high toxicity levels collectively give a hint but did not represent a specific point where one could say at the point x there is a risk. Kindly use the propit analysis.
See the reply in comment 1.

Reviewer 3 Report
Review for the paper "The Influence of the Toxic Dinoflagellate Alexandrium minutum, grown under different N/P ratios, on the Marine Copepod Acartia tonsa" by anonymous authors submitted to "Toxins".
General comment.
Microalgae have been revealed to play several important roles in marine pelagic food webs, e.g. as primary producers thus controlling the efficiency of carbon cycling in marine ecosystems. Moreover, some taxa can release toxins that may have a great impact on the higher trophic levels from zooplankton to fish, mammals, and birds. Copepods are responsible for many ecological and biogeochemical functions and the alteration of the populations in response to global natural changes may have the potential to alter global biogeochemical cycles, ecosystem functioning and predator−prey interactions, causing trophic mismatch. Harmful algae blooms can regulate population dynamics of coastal copepods in many regions. Therefore, studying various aspects related to this problem is considered to be one of the most challenging issues in aquatic ecology. The copepods of the genus Acartia are common members of coastal waters worldwide and these are good experimental organisms to study effects of various environmental stressors. The authors revealed that toxins released by the toxic dinoflagellate Alexandrium minutum negatively affected the egg and faecal pellet production as well as the feeding behavior of A. tonsa. The paper is well structured. The introduction provides relevant background, methods appear to be valid for this kind of research. Results are presented in a good way. Statistical methodology seems to be adequate. Discussion provides an interpretation based on the main findings. The paper may be interesting to a broad audience. I suggest minor changes to improve the ms.
Specific remarks.
L88, 166. A. tonsa must be in Italics.
L91-92. Alexandrium minutum must be in Italics.
L107. Alexandrium minutum must be in Italics.
L108, 237. Consider replacing "In the different A. minutum concentrations" with "At the different A. minutum concentrations".
L143. Consider replacing "500." with "500,".
L154-156. Unclear sentence. According to p-level (<0.001) there must be significant differences between toxicity levels although the authors noted no significant differences. Please, check.
L394. ‘d–1'. '-1' must be in the upper index.
Discussion. Provide a short section to describe the ecological application of your results focusing on the possible role of HAB in regulating carbon cycles in costal ecosystems and in causing changes in zooplankton populations.
Author Response
General comment.
Microalgae have been revealed to play several important roles in marine pelagic food webs, e.g. as primary producers thus controlling the efficiency of carbon cycling in marine ecosystems. Moreover, some taxa can release toxins that may have a great impact on the higher trophic levels from zooplankton to fish, mammals, and birds. Copepods are responsible for many ecological and biogeochemical functions and the alteration of the populations in response to global natural changes may have the potential to alter global biogeochemical cycles, ecosystem functioning and predator−prey interactions, causing trophic mismatch. Harmful algae blooms can regulate population dynamics of coastal copepods in many regions. Therefore, studying various aspects related to this problem is considered to be one of the most challenging issues in aquatic ecology. The copepods of the genus Acartia are common members of coastal waters worldwide and these are good experimental organisms to study effects of various environmental stressors. The authors revealed that toxins released by the toxic dinoflagellate Alexandrium minutum negatively affected the egg and faecal pellet production as well as the feeding behavior of A. tonsa. The paper is well structured. The introduction provides relevant background, methods appear to be valid for this kind of research. Results are presented in a good way. Statistical methodology seems to be adequate. Discussion provides an interpretation based on the main findings. The paper may be interesting to a broad audience. I suggest minor changes to improve the ms.
Specific remarks.
L88, 166. A. tonsa must be in Italics.
It was adjusted.
L91-92. Alexandrium minutum must be in Italics.
It was adjusted.
L107. Alexandrium minutum must be in Italics.
It was adjusted.
L108, 237. Consider replacing "In the different A. minutum concentrations" with "At the different A. minutum concentrations".
It was adjusted.
L143. Consider replacing "500." with "500,".
It was adjusted.
L154-156. Unclear sentence. According to p-level (<0.001) there must be significant differences between toxicity levels although the authors noted no significant differences. Please, check.
It was corrected (<0.1).
L394. ‘d–1'. '-1' must be in the upper index.
It was adjusted.
Discussion. Provide a short section to describe the ecological application of your results focusing on the possible role of HAB in regulating carbon cycles in costal ecosystems and in causing changes in zooplankton populations.
Two new sentences are now included at the end of the Discussion.

Round 2
Reviewer 2 Report
Dear Drs
Happy day.
Many thanks
You have improve the paper.
With my best wishes
Amro Amara